# Phase 1 Study to Evaluate the Safety of Reducing the Prophylactic Dose of Dexamethasone around Docetaxel Infusion in Patients with Prostate and Breast Cancer

**DOI:** 10.3390/cancers15061691

**Published:** 2023-03-09

**Authors:** Rieneke T. Lugtenberg, Stefanie de Groot, Danny Houtsma, Vincent O. Dezentjé, Annelie J. E. Vulink, Maarten J. Fischer, Johanneke E. A. Portielje, Jacobus J. M. van der Hoeven, Hans Gelderblom, Hanno Pijl, Judith R. Kroep

**Affiliations:** 1Department of Medical Oncology, Leiden University Medical Center, 2333 ZA Leiden, The Netherlands; 2Department of Medical Oncology, Haga Hospital, 2545 AA Den Haag, The Netherlands; 3Department of Medical Oncology, Reinier de Graaf Hospital, 2625 AD Delft, The Netherlands; 4Department of Pharmacy & Pharmacology, Antoni van Leeuwenhoek-The Netherlands Cancer Institute, 1066 CX Amsterdam, The Netherlands; 5Department of Medical Psychology, Leiden University Medical Center, 2333 ZA Leiden, The Netherlands; 6Department of Endocrinology, Leiden University Medical Center, 2333 ZA Leiden, The Netherlands

**Keywords:** prostate cancer, breast cancer, dexamethasone, docetaxel, toxicity, hypersensitivity reactions, fluid retention syndrome, phase 1 study

## Abstract

**Simple Summary:**

Docetaxel has been approved as an anti-cancer agent in 1995. High rates of hypersensitivity reactions (HSR) and fluid retention were observed when this agent was first introduced. The use of high dose systemic corticosteroids around docetaxel infusion appeared to decrease the incidence of HSR and fluid retention and has been applied in daily practice ever since. However, there is little evidence that supports this high dose of dexamethasone. Furthermore, the application of high-dosed corticosteroids can lead to undesirable adverse effects. In this phase 1 study, we aim to evaluate the impact of reducing the dose of dexamethasone as an adjunct to docetaxel on the incidence of HSR and fluid retention in patients with prostate or breast cancer.

**Abstract:**

Background: There is little evidence that supports the registered high dose of dexamethasone used around docetaxel. However, this high dose is associated with considerable side effects. This study evaluates the feasibility of reducing the prophylactic oral dosage of dexamethasone around docetaxel infusion. Patients and methods: Eligible patients had a histologically confirmed diagnosis of prostate or breast cancer and had received at least three cycles of docetaxel as monotherapy or combination therapy. Prophylactic dexamethasone around docetaxel infusion was administered in a de-escalating order per cohort of patients. Primary endpoint was the occurrence of grade III/IV fluid retention and hypersensitivity reactions (HSRs). Results: Of the 46 enrolled patients, 39 were evaluable (prostate cancer (*n* = 25), breast cancer (*n* = 14). In patients with prostate cancer, the dosage of dexamethasone was reduced to a single dose of 4 mg; in patients with breast cancer, the dosage was reduced to a 3-day schedule of 4 mg–8 mg–4 mg once daily, after which no further reduction has been tested. None of the 39 patients developed grade III/IV fluid retention or HSR. One patient (2.6%) had a grade 1 HSR, and there were six patients (15.4%) with grade I or II edema. There were no differences in quality of life (QoL) between cohorts. Conclusions: It seems that the prophylactic dose of dexamethasone around docetaxel infusion can be safely reduced with respect to the occurrence of grade III/IV HSRs or the fluid retention syndrome.

## 1. Introduction

Docetaxel—a semisynthetic analog of paclitaxel, causing cell-cycle arrest and apoptosis through interference with microtubular function—has been registered as an anticancer agent in 1995 [1,2]. In the early clinical trials, high rates of hypersensitivity reactions (HSRs) were observed during taxane infusion. The occurrence of HSRs decreased to less than 10% after prophylactic medication with H1 and H2 antihistamines and when systemic corticosteroids became part of cancer treatment protocols [3,4]. Additionally, a fluid retention syndrome, characterized by weight gain, edema, and pleural effusion, was observed after docetaxel administration, which resulted in treatment discontinuation in 30–70% of patients. It was found to be a cumulative, dose-limiting, and slowly reversible toxicity [5,6,7,8,9,10,11]. Corticosteroids, first given to prevent HSR, appeared to prevent fluid retention associated with docetaxel as well [5,6,7,8].

Docetaxel is approved for the treatment of breast cancer with the concomitant use of a 3-day schedule of dexamethasone 8 mg, bi-daily (bid), starting on the day before chemotherapy, with the purpose to decrease the severity of fluid retention and HSRs [12]. The commonly used dosage of docetaxel in treatment schedules for breast cancer is 100 mg/m^2^. For the treatment of prostate cancer, a lower dosage of 75 mg/m^2^ docetaxel is registered. Due to this lower dosage and the concurrent use of low-dose prednisone in prostate cancer treatment schedules, this is accompanied by another prophylactic regimen: 3 times of 8 mg of dexamethasone on the day of docetaxel infusion [12]. There is little evidence that supports the high doses of dexamethasone used in both schedules. For example, the use of the 3-day 8 mg bid schedule for the docetaxel dosage of 100 mg/m^2^ is based on a conference abstract [13]. Dexamethasone potentially has severe side effects and can evoke manifestations of diabetes mellitus, weight gain, gastro-esophageal reflux disease, personality changes, irritability, insomnia, agitation, euphoria, mania, and mood swings [14,15,16]. The use of corticosteroids can induce immunosuppression with an increased risk of infection—the risk is even higher when myelosuppressive chemotherapy is given simultaneously [16]. In addition, there is increasing evidence that the occurrence of diabetes, causing high values of glucose and insulin, can worsen the prognosis of cancer patients [17,18]. Data from pre-clinical and clinical studies suggest that corticosteroids can induce treatment resistance in solid tumors [19]. In recent years, it has been established that glucocorticoid receptors (GRs) may be involved in the development of castration-resistant prostate cancer (CRPC) [20,21]. The upregulation of the GR may drive tumor proliferation and possibly lead to resistance to antiandrogen therapies [22]. As a consequence, the use of dexamethasone and other corticosteroids may contribute to tumor progression in prostate cancer [23,24]. Finally, dexamethasone is a CYP3A4 inducer, which might increase docetaxel clearance [25,26]. Thus, there is a need to re-evaluate the optimal dose of prophylactic dexamethasone.

In this phase 1 study, we evaluated the impact of reducing the dose of dexamethasone as an adjunct to docetaxel on HSR and fluid retention in patients with prostate or breast cancer.

## 2. Materials and Methods

### 2.1. Study Design

This study is a multicenter, open label, dose-de-escalating, non-randomized phase 1 study. Patients received docetaxel infusion every 3 weeks for a minimum of 3 cycles, depending on the regimen, until progressive disease or unacceptable toxicity. Prophylactic dexamethasone co-medication was administered in a de-escalating order (Table 1) per cohort (Table 1). Six patients were enrolled per dose level initially. Each patient within a cohort received the same dose of dexamethasone in every subsequent cycle. The last patients of a cohort were observed for 2 cycles of docetaxel treatment before accrual to the next lower dose level was started. Patients were replaced within a cohort if they left the study within 3 weeks for reasons other than toxicity. If no grade III/IV HSR or fluid retention reaction occurred in the six patients within on cohort, the next cohort was treated with the next dose level. If one grade III/IV HSR or fluid retention reaction occurred in one of the six patients within one cohort, then three additional patients were treated at that dose level. If there were no additional grade III/IV HSR or fluid retention in the additional 3 patients, accrual to the next lower dose level was started. If a grade III/IV HSR or fluid retention occurred in at least 2/6 or 2/9 patients, that dose was not considered as safe. Each patient within a cohort received the same dose of dexamethasone in every subsequent cycle. Patients were replaced within a cohort if they left the study within 3 weeks for reasons other than toxicity. Initially, for the breast cancer group, 3 additional cohorts were planned (cohort 4: day 0: 8 mg and day 1: 4 mg; cohort 5: day 0: 8 mg; and cohort 6 day 0: 4 mg). However, the inclusion of patients with breast cancer was stopped after cohort 3, as inclusion was falling behind due to an increase in the use of weekly paclitaxel instead of docetaxel in this group of patients.

### 2.2. Patient

Eligible patients had histologically confirmed the diagnosis of prostate cancer or breast cancer and a treatment plan with a minimum of 3 cycles of docetaxel monotherapy or combination therapy. Patients with prostate cancer can be treated with or without bi-daily (bid) 5 mg prednisone continuously. Patients had an adequate bone marrow function (i.e., white blood counts > 3.0 × 10^9^/L, absolute neutrophil count ≥ 1.5 × 10^9^/L, and platelet count ≥ 100 × 10^9^/L) no signs of liver damage (i.e., bilirubin ≤ 1.5 × upper limit of normal (UNL) range, ALAT and/or ASAT ≤ 2.5 × UNL, and Alkaline Phosphatase ≤ 5 × UNL), adequate renal function (i.e., calculated creatinine clearance ≥ 50 mL/min), a WHO performance status of 0–2, age ≥ 18 years, a survival expectation of >3 months, an absence of diabetes mellitus, an absence of steroid use for other conditions, an absence of pregnancy or current lactation, an absence of existing edema, and written informed consent. Patients were excluded if they had a known hypersensitivity for docetaxel, paclitaxel, other chemotherapeutic agents, products containing polysorbate 80, or an earlier experience of anaphylaxis for food, insect bites, medication, or another foreign substance.

### 2.3. Endpoint

The primary endpoint was HSR or fluid retention syndrome grade III/IV. During each cycle, toxicity was documented by the physician and graded according to the Common Terminology Criteria for Adverse Events version 4.03 (CTCAE v.4.03) [26].

### 2.4. Quality of Life

Quality of life was assessed with the European Organization for Research and Treatment of Cancer—Core Quality of Life Questionnaire (EORTC-QLQ C30) before the start of treatment, after 3 cycles and after 6 cycles of docetaxel. This 30-item test comprises one global health scale, five function scales (physical, emotional, cognitive, social, and role), three symptom scales (fatigue, nausea, and pain), and six single items. All scores were transformed to a 0–100 scale.

### 2.5. Statistical Analysis

Other trials with the monotherapy of docetaxel showed that 5–6% of patients experienced grade III/IV fluid retention [27,28,29] and 2.5% of patients experienced grade III/IV HSR [27] with concomitant prophylactic high-dose dexamethasone. Therefore, the combined endpoint of grade III/IV fluid retention or HSR was expected to occur in approximately 8% of patients. We deemed doubling in the occurrence of grade III/IV fluid retention and HSR to be acceptable. Thus, one out of six patients within the one-dose cohort who experienced grade III/IV fluid retention or HSR would be above the maximal accepted doubling of side effects. In that case, three additional patients were treated at the same dose level. If there was no further occurrence of grade III/IV fluid retention or HSR within that cohort, the toxicity remained under the accepted doubling (one out of nine patients, 11%). Grade III/IV HSR or fluid retention occurred in at least 2/6 or 2/9 patients, thus, the estimation of occurrence of grade III/IV fluid retention or HSR was unacceptably high in the study, with 33% or 22%, respectively. Therefore, the last high dose level of dexamethasone will be the recommended dose for a phase III study. The differences between cohorts on serum levels for glucose, insulin, and IGF-1 levels were tested with the Mann–Whitney U test. The differences between cohorts on the different QoL scales were estimated using linear mixed models, with an unstructured covariance matrix including cohorts, time, and the interaction between cohorts and time. For each scale, all scores over time were used as the dependent outcome in the models. All tests were two-tailed with a significance level of *p* < 0.05. All data were analyzed using IBM SPSS Statistics for Windows (Version 25.0. Armonk, NY, USA: IBM Corp).

## 3. Results

### 3.1. Patient Characteristics

From April 2016 to June 2020, 28 patients with prostate cancer and 18 patients with breast cancer from three participating Dutch centers were included. Patient characteristics are shown in Table 2. A total of 39/46 patients were evaluable for toxicity. Three patients used the normal dosage of dexamethasone and violated study protocol. One patient declined to participate in the study and withdrew informed consent, two patients stopped the docetaxel treatment early, and one patient switched to paclitaxel treatment (Figure 1).

### 3.2. Toxicity

The percentage of patients in whom a hypersensitivity reaction (HSR) and/or fluid retention was observed is shown in Table 3. No grade III/IV toxicity occurred in any of the cohorts. One patient developed a grade I HSR. After this mild reaction, the patient decided to use the normal dosage of dexamethasone in the consequent cycles and left the study. Six patients (15%) had grade I or II fluid retention, consisting of mild-to-moderate edema; no pleural effusions were observed. Febrile neutropenia, nausea, and hyperglycemia occurred in up to 33% (Table 4).

No differences were found in the median levels of glucose, insulin, or IGF-1 between cohorts (Appendix A).

### 3.3. Quality of Life

The QLQ-C30 was completed in all 24 patients in the prostate cancer cohort and 10/14 of the patients in the breast cancer cohort before the start of docetaxel treatment. The scores were comparable between the different cohorts (Appendix A). Some of the scores deteriorated similarly during docetaxel treatment, other scores had different patterns over time. However, there were no differences between cohorts on any of the EORTC-QLQ C30 scales or items during the tree time points (Appendix A).

## 4. Discussion

This phase 1 dose-finding study evaluates the feasibility of reducing the optimal dose of dexamethasone comedication for docetaxel treatment and demonstrates the feasibility of reducing it. Our results show that reducing the dose of dexamethasone to a single dose of 4 mg before docetaxel administration in patients with prostate cancer (docetaxel dosage of 75 mg/m^2^)—or to at least to 4 mg on day −1, 8 mg on day 0, and 4 mg on day 1 in patients with breast cancer (docetaxel dosage of 100 mg/m^2^)—does not increase the incidence of hypersensitivity reactions (HSRs) or fluid retention syndrome. Furthermore, there were no differences in perceived QoL, nor did patients experience increased nausea, fatigue, or loss of appetite. High-dose dexamethasone may be associated with severe side effects such as metabolic changes, gastro-intestinal conditions, and behavior change [14,15,16]. However, the use of dexamethasone around docetaxel treatment is mostly short-term and HSRs and fluid retention syndrome can be life-threatening, so tapering should be carried out cautiously. Therefore, reducing or withholding dexamethasone premedication is nevertheless desirable, especially when evidence to support the prescription of high-dose dexamethasone co-medication is lacking. Upfront therapy with six courses of docetaxel in addition to androgen deprivation therapy (ADT) has been shown to improve overall survival in patients with metastatic hormone-sensitive prostate cancer (mHSPC) and has become the standard of care [30,31]. In the CHAARTED trial, the concomitant use of prednisone was not mandatory; therefore, nowadays, many patients with prostate cancer receive six cycles of docetaxel 75 mg/m^2^ without bi-daily 5 mg of prednisone. We show that even in this group of patients, dexamethasone can be safely reduced to a single dose of 4 mg before docetaxel infusion. Thus, it seems that patients can receive six courses of docetaxel treatment safely with a single dose of 4 mg dexamethasone premedication for each course, instead of three times of 8 mg, even without chronic prednisone use (cohort 3B prostate cancer). This supports a significant reduction in corticosteroid use with a beneficial reduction in associated adverse effects.

Our findings were consistent with the results of a few previous studies, in which lower doses of dexamethasone during docetaxel treatment were investigated [32,33,34,35,36]. None of these studies reported an increase in the incidence of HSRs or fluid retention if dexamethasone was given in a lower dose. Chen et al. safely reduced the dose of the recommended dexamethasone as an adjunct of docetaxel (dosage 70 mg/m^2^) in patients with head and neck cancer from 45 to 11 mg without an increase in severe HSRs or edema [33]. Accordingly, other studies reported no differences in HSRs or fluid retention after a single dose of dexamethasone IV before docetaxel administration in weekly or 3-weekly treatment schedules for the treatment of various solid tumors [32,35], nor after a 3-day regimen with a lower dose of dexamethasone (4.5 mg once a day) in comparison with the standard regimen [33]. To our knowledge, this is the first prospective study in which a single dose of oral dexamethasone premedication before docetaxel treatment has been investigated.

Chemotherapy-induced (febrile) neutropenia and infections can be life-threatening and dose-limiting adverse events. High-dose dexamethasone might increase this risk even further because it causes lymphopenia and has an immunosuppressive effect. Furthermore, steroid-induced hyperglycemia may contribute to an increased risk of infection as well. None of our patients in the cohorts with the lowest dose of dexamethasone had febrile neutropenia and only one patient had grade 1 hyperglycemia, while in the first two cohorts, up to one third of patients had grade 1–2 hyperglycemia and four patients (17%) were hospitalized due to febrile neutropenia. We realize that the number of patients in each cohort is limited. Nevertheless, we believe our data do support our hypothesis and show a clear trend in favor of lower dosages of dexamethasone. Moreover, our findings are in line with similar observations reported previously. For example, in the study of Kang et al., less infectious complications were observed in patients treated with a single dose of 10 mg of dexamethasone IV in comparison with patients who received the oral bid dosage of 4 mg of dexamethasone for 2 days in addition to the 10 mg of IV before docetaxel administration [32]. The lower risk of infectious complications was also observed in studies using lower doses of dexamethasone as anti-emetic regimens [37].

The occurrence of the fluid retention syndrome has a strong correlation with the cumulative dose of administered docetaxel. In phase II studies, before the introduction of corticosteroid premedication regimens, the median cumulative dose at the onset of fluid retention was between 300 and 400 mg/m^2^ [38]. Nowadays, most patients with prostate or breast cancer will not receive more than 400–450 mg/m^2^ of docetaxel. The mean dosages in our study were 412 mg/m^2^ (range 75–645) in patients with prostate cancer and a mean of 394 mg/m^2^ (range 100–600) in patients with breast cancer. In contrast, treatment regimens given in the early phase quite often entailed dosages up to 600–700 mg/m^2^. Therefore, a severe fluid retention syndrome is less likely to occur in current clinical practice. Two patients with breast cancer (14%) and four patients with prostate cancer (16%) developed mild-to-moderate edema. However, this may also have been an adverse effect of the dexamethasone premedication, as edema also occurred in the first cohorts that used higher dexamethasone dosages. This finding is all the more reason to critically assess the necessity of (high-dose) dexamethasone as an adjunct to chemotherapy.

Patients are most at risk of developing hypersensitivity reactions (HSRs) during the first or second infusion, and life-threating reactions are very rare [39]. In our study, none of the patients developed a severe HRS, despite using lower dosages of dexamethasone premedication, and the only non-severe HRS (grade 1) occurred during the first chemotherapy cycle. Parinyanitakul et al. and Barrosa-Sousa et al. showed that in patients with early breast cancer treated with paclitaxel, dexamethasone premedication could be withheld safely if patients did not experience HSR in response to the two previous cycles [40,41]. It is conceivable that the same applies for dexamethasone if prescribed as an adjunct to docetaxel administration.

Our trial was limited by the small number of patients enrolled in the study, especially the number of patients with breast cancer. We initially planned to include at least 36 patients in six cohorts of patients with breast cancer. However, during the study period, breast cancer treatment protocols in the Netherlands changed rapidly to include weekly paclitaxel instead of 3-weekly docetaxel because of a more favorable side-effect profile. As a consequence, we prematurely stopped the inclusion of patients with breast cancer. In the 14 patients we included, it appeared safe to reduce the dexamethasone dose by more than half (from a cumulative dose of 48 mg to 16 mg), as it did not increase the incidence of HSRs or fluid retention. In view of the results in patients with prostate cancer, it is conceivable that the dexamethasone dose can be reduced even further. Future studies with a larger number of patients could possibly establish the “median effective dose (ED_50_)” of prophylactic dexamethasone for different docetaxel regimens. The lower dosage of dexamethasone could indeed be worthwhile in view of its potential side effects and the growing evidence that hyperglycemia and hyperinsulinemia (the metabolic effects of dexamethasone) may be associated with poorer outcomes in patients with cancer [42,43,44]. However, since dexamethasone is also used for its effective antiemetic properties, it may not be appropriate to omit it completely. Nevertheless, recent clinical trials have demonstrated the benefit of using prednisone, instead of dexamethasone, in conjunction with docetaxel for the treatment of advanced prostate cancer [45]. Furthermore, Tanaka et al. showed that docetaxel combined with 0.5 mg of dexamethasone orally twice a day results in a PSA response and good survival efficacy in castration-resistant prostate cancer [46,47], whereas other trials in patients with metastatic castration-resistant prostate cancer displayed benefits regarding switching corticosteroid from prednisone to dexamethasone after progression in abiraterone acetate [48,49]. This implies that particular patients with advanced prostate cancer might benefit from the use of corticosteroids. As dexamethasone is also used for its effective antiemetic properties, it may not be appropriate to omit it completely in docetaxel treatment.

## 5. Conclusions

Our study strongly suggests that the prophylactic dose of dexamethasone as an adjunct of docetaxel infusion may be safely reduced without increasing the risk of hypersensitivity reactions or fluid retention syndrome. However, in view of the small number of patients included, larger studies are required to confirm our results.

## Figures and Tables

**Figure 1 cancers-15-01691-f001:**
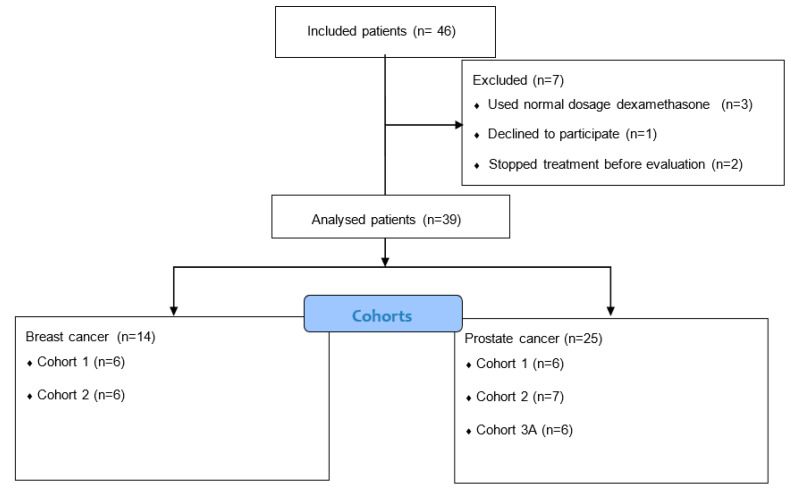
Consort diagram.

**Table 1 cancers-15-01691-t001:** Dose levels of dexamethasone per cohort.

Breast Cancer (Docetaxel Dosage 75–100 mg/m^2^)						
	Day −1		Day 0		Day 1	
Cohort 1	8 mg	4 mg	8 mg	4 mg	8 mg	4 mg
Cohort 2	8 mg		8 mg		8 mg	
Cohort 3	4 mg		8 mg		4 mg	
Prostate cancer (Docetaxel dosage 75 mg/m^2^)						
	Day −1		Day 0			
Cohort 1	8 mg		8 mg			
Cohort 2	-		8 mg			
Cohort 3A	-		4 mg, with 2dd 5 mg prednisone continuously			
Cohort 3B	-		4 mg, without 2dd 5 mg prednisone continuously			

**Table 2 cancers-15-01691-t002:** Patient characteristics.

	Prostate Cancer (N = 28)	Breast Cancer (N = 18)
Median Age (range), Years	69.5 (55–80)	54 (34–67)
Median Body Mass Index (range), kg/m^2^	26.9 (21.3–32.7)	26.4 (18.6–40.4)
WHO Status		
Grade 0	15 (54%)	12 (67%)
Grade 1	6 (21%)	3 (17%)
Grade 2	1 (4%)	0 (0%)
Missing	6 (21%)	3 (17%)
Stage		
I	0 (0%)	3 (17%)
II	0 (0%)	12 (67%)
III	0 (0%)	1 (6%)
IV	28 (100%)	2 (11%)
Chemotherapy Regimen		
(Neo)adjuvant	0 (0%)	16 (89%)
Palliative	28 (100%)	2 (11%)
Docetaxel Dosage		
75 mg/m^2^	28 (100%)	2 (11%)
100 mg/m^2^	0 (0%)	16 (89%)
Treatment		
Monotherapy Docetaxel	28 (100%)	14 (78%)
Combination Chemotherapy	0 (0%)	4 (22%)

Frequencies (%), some percentages may not total 100 because of rounding.

**Table 3 cancers-15-01691-t003:** The occurrence of hypersensitivity reactions (HSR) and fluid retention (More details can be seen in Appendix A).

**Breast Cancer**	**Number of Patients Evaluated**	**Median Cycles of Docetaxel (Range)**	**Mean Cumulative Dose of Docetaxel (Range)**	**HSR** **(Any Grade)**	**HSR** **(Grade III/IV)**	**Fluid Retention * (Any Grade)**	**Fluid Retention (Grade III/IV)**
Cohort 1	6	4 (4)	351 mg/m^2^ (325–400)	0	0	1/6 (grade I)	0
Cohort 2	6	4 (1–4)	307 mg/m^2^ (100–400)	0	0	1/6 (grade II)	0
Cohort 3	2	6 (6)	525 mg/m^2^ (450–600)	0	0	0	0
**Prostate Cancer**	**Number of Patients Evaluated**	**Median Cycles of Docetaxel (Range)**	**Mean Cumulative Dose of Docetaxel (Range)**	**HSR** **(Any Grade)**	**HSR** **(Grade III/IV)**	**Fluid Retention * (Any Grade)**	**Fluid Retention (Grade III/IV)**
Cohort 1	6	6 (1–9)	445 mg/m^2^ (75–645)	0	0	1/6 (grade II)	0
Cohort 2	7	6 (5–6)	407 mg/m^2^ (225–450)	1/6 (grade I) **	0	0	0
Cohort 3A	6	6 (2–6)	372 mg/m^2^ (150–450)	0	0	1/6 (grade I/II)	0
Cohort 3B	6	6 (5–6)	434 mg/m^2^ (356–450)	0	0	2/6 (grade I/II)	0

Toxicity graded according to the Common Terminology Criteria for Adverse Events version 4.03 (CTCAE v.4.03); * all fluid retention reported in the different cohorts consisted of mild-to-moderate oedema, no pleural effusions were seen; ** after a grade 1 HSR at the first cycle of docetaxel, this patient decided to use the normal dosage of dexamethasone in the consequent cycles and left the study.

**Table 4 cancers-15-01691-t004:** Other toxicity of interest.

	Febrile Neutropenia	Nausea	Hyperglycaemia
Cohorts 1	3/12 (grade IV) *	5/12 (grade I–II)	4/12 (grade I–II)
Cohorts 2	1/13 (grade IV) *	5/13 (grade I)	3/13 (grade I)
Cohorts 3	0/14	2/14 (grade I–II)	1/14 (grade I)

Toxicity was graded according to the Common Terminology Criteria for Adverse Events version 4.03 (CTCAE v.4.03); * 3 patients in cohort 1 (2 prostate cancer, 1 breast cancer) and 1 patient in cohort 2 (breast cancer) were hospitalized due to febrile neutropenia.

## Data Availability

The authors confirm they have full control of all primary data and agree to allow the journal to review their data if requested. The datasets generated and/or analyzed during the current study are not publicly available as sharing is not explicitly covered by patient consent.

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
