# Peer review of "Phase 1 Study to Evaluate the Safety of Reducing the Prophylactic Dose of Dexamethasone around Docetaxel Infusion in Patients with Prostate and Breast Cancer"

_cancers, 2023, doi:10.3390/cancers15061691_

Round 1
Reviewer 1 Report
1. Please check the English grammars and punctuation use. For example, Line 26, “In this phase 1 study” add (,) after ‘phase 1 study’. Or Line 143, “From April 2016 to June 2020” please use comma after ‘2020’.
2. Please separate the value and unit. Such as “4mg” consider to revise to “4 mg” Please do so to the entire manuscript.
3. Discussion in the introduction is balance. Maybe check for grammatical and punctuation use errors.
4. Please provide the study design in Materials and Methods section
5. 2.3 “Toxicity” maybe can be changed to “Endpoint”?
6. Other than stated, the methods appears to be clear enough. But maybe, please check the technical writing of Table 1. I don’t agree with the current presentation of the table especially its column headings.
7. “2,5%” check again the decimal separator. Kindly use (.) as international standard. The same thing with ” 69,5” in Table 2.
8. “significance level of 0.05” you mean p<0.05?
9. “total 100 because of rounding.” What is rounding?
10. Please give definition for ‘gr’. It’s supposed to indicate grade (?) Kindly explain in the text or table footnotes.
11. Please present the p-value for all data that have been analyzed statistically. Comparisons should be made between those receiving normal dose (the higher one) and those receiving lower dose.
12. Is it possible to calculate the “median effective dose (ED50)” of prophylactic dexamethasone? Maybe for the next investigations? Kindly elaborate in the discussion.
Author Response
Response to reviewer 1
- Please check the English grammars and punctuation use. For example, Line 26, “In this phase 1 study” add (,) after ‘phase 1 study’. Or Line 143, “From April 2016 to June 2020” please use comma after ‘2020’.
Thank you so much, we have checked the English grammars and punctuation use in the manuscript and have adopted the above suggestions.
- Please separate the value and unit. Such as “4mg” consider to revise to “4 mg” Please do so to the entire manuscript.
It has been separated in the entire manuscript and in tables and figures.
- Discussion in the introduction is balance. Maybe check for grammatical and punctuation use errors.
The grammatical and punctuation use in the introduction and discussion section has been checked and adjusted where necessary.
- Please provide the study design in Materials and Methodssection
The study design has been added in the methods section.
- 2.3 “Toxicity” maybe can be changed to “Endpoint”?
2.3 “Toxicity” has been changed to “Endpoint”
- Other than stated, the methods appears to be clear enough. But maybe, please check the technical writing of Table 1. I don’t agree with the current presentation of the table especially its column headings
Table 1 has been adjusted as followed, to make it more clear:
Table 1. De-escalating dose levels dexamethasone per cohort
|
Breast cancer (Docetaxel dosage 75-100 mg/m2)
|
|||||||
|
|
Day before chemotherapy
|
Day of chemotherapy |
Day after chemotherapy |
||||
|
Cohort 1 |
8 mg |
4mg |
8 mg |
4 mg |
8 mg |
4 mg |
|
|
Cohort 2 |
8 mg |
8 mg |
8 mg |
||||
|
Cohort 3 |
4 mg |
8 mg |
4 mg
|
||||
|
Prostate cancer (Docetaxel dosage 75 mg/m2)
|
|||||||
|
|
Day before chemotherapy
|
Day of chemotherapy |
|||||
|
Cohort 1 |
8 mg |
8 mg |
|||||
|
Cohort 2 |
- |
8 mg |
|||||
|
Cohort 3A |
- |
4 mg, with 2dd 5 mg prednisone continuously |
|||||
|
Cohort 3B |
- |
4 mg, without 2dd 5 mg prednisone continuously
|
|||||
- “2,5%” check again the decimal separator. Kindly use (.) as international standard. The same thing with ” 69,5” in Table 2.
Thank you for pointing out this inconsistence. It has been adapted in the manuscript and tables.
- “significance level of 0.05” you mean p<0.05?
Indeed, we meant a p<0.05. This has been added in the statistical analysis section.
- “total 100 because of rounding.” What is rounding?
Some percentages may not total 100 because of rounding to the nearest whole number. This explanation has been added in the footnote of the table.
- Please give definition for ‘gr’. It’s supposed to indicate grade (?) Kindly explain in the text or table footnotes.
“gr” has been changed into grade in table 3 and 4
- Please present the p-value for all data that have been analyzed statistically. Comparisons should be made between those receiving normal dose (the higher one) and those receiving lower dose.
Because it is a phase 1 study with only a small number of patients this trial we chose to provide only descriptive results of febrile neutropenia, nausea and hyperglycaemia, this has not been statistically analyzed. The analysis of the glucose, insulin and IGF-1 levels (supplementary materials) have been adapted and are now compared with the first dose level.
- Is it possible to calculate the “median effective dose (ED50)” of prophylactic dexamethasone? Maybe for the next investigations? Kindly elaborate in the discussion.
Thank you for this suggestion. It has been added in the discussion as a suggestion for next investigations.

Reviewer 2 Report
The authors address an important but not novel issue in the field, to reduce the prophylactic dose of dexamethasone in prostate cancer and breast cancer patients receiving docetaxel to minimize the undesirable adverse effects of corticosteroids. This issue is of particular importance in the context of corticosteroids contributing to therapy resistance in solid tumors by diminishing taxane-based anti-tumors effects.
Major points:
1. The key limitation of the study was the very low number of patients enrolled, with a highly descriptive analysis of the results. This prevented a robust statistical analysis, which precluded reaching sound scientific conclusions.
2. It is not clear what cancer types were represented in the prostate cancer cohort (i.e. localized, metastatic non-castration resistant, metastatic castration-resistant, etc) and in the breast cancer cohort (luminal, receptor status, TBNC, etc). The administration of dexamethasone together with docetaxel may have differential effects on the patients depending on the disease type.
3. The rationale provided in the Introduction for diminishing the dexamethasone dose in the context of chemotherapy resistance and tumor progression cites references 20-23, which are 10-20 years old. During the past 10 years, there has been a wealth of research on the effects of dexamethasone and its ligand glucocorticoid receptor on therapy resistance in various solid tumors including breast, ovarian, and prostate cancers. In the case of prostate cancer, glucocorticoid receptor signaling has been recently linked to both chemotherapy resistance and anti-androgen therapy resistance. This newer literature is not discussed.
4. Did the authors examine the patients for the effects of reduced dexamethasone dose on symptoms other than fluid retention and hypersensitivity reactions?
5. Recent clinical trials have demonstrated the benefit of using prednisone or prednisolone, instead of dexamethasone, in conjunction with docetaxel or abiraterone for the treatment of advanced prostate cancer. How would the results presented in this manuscript with a small cohort of patients impact the current treatment of advanced prostate cancer in light of the several recent clinical trials evaluating the benefits of prednisone compared to dexamethasone?
6. The recent multi-center studies by Tanaka et al (Int J Urol 2019, Japan J Clin Oncology 2017) in larger patient cohort studied the safety of docetaxel-chemotherapy combined with low-dose dexamethasone (0.5-1mg) and demonstrated excellent safety with a survival benefit for castration-resistant prostate cancer patients. The authors did not compare their results with a reduced dose of dexamethasone (4 mg) to those of Tanaka et al. Further, the studies by Tanaka et al were not cited.
7. How did the study account for patients using corticosteroid treatment for other conditions while enrolled in this clinical study?
Minor points
1. Lines 112 – 113 have different font types
2. In table 3, “Number ofevaluablepatients” should indicate "number of patients evaluated"
3. The link on reference 12 takes to a page that no longer exists.
Author Response
Response to reviewer 2
Major points:
- The key limitation of the study was the very low number of patients enrolled, with a highly descriptive analysis of the results. This prevented a robust statistical analysis, which precluded reaching sound scientific conclusions.
You are right that it is a small study. In our enthusiasm we may have drawn conclusions too strongly. The conclusion in the abstract has been adjusted. Furthermore, in the discussion we emphasized this even more to add a more nuance. We already stated in the conclusion that because of the small number of patients, larger studies are required to confirm our results.
- It is not clear what cancer types were represented in the prostate cancer cohort (i.e. localized, metastatic non-castration resistant, metastatic castration-resistant, etc) and in the breast cancer cohort (luminal, receptor status, TBNC, etc). The administration of dexamethasone together with docetaxel may have differential effects on the patients depending on the disease type.
On request of another reviewer as well, table 3 can be replaced with a new table we made with more detailed information for each included patient
- The rationale provided in the Introduction for diminishing the dexamethasone dose in the context of chemotherapy resistance and tumor progression cites references 20-23, which are 10-20 years old. During the past 10 years, there has been a wealth of research on the effects of dexamethasone and its ligand glucocorticoid receptor on therapy resistance in various solid tumors including breast, ovarian, and prostate cancers. In the case of prostate cancer, glucocorticoid receptor signaling has been recently linked to both chemotherapy resistance and anti-androgen therapy resistance. This newer literature is not discussed.
Thank you for this suggestion. This is now added in the introduction section and newer references have been included.
- Did the authors examine the patients for the effects of reduced dexamethasone dose on symptoms other than fluid retention and hypersensitivity reactions?
Because it is a phase 1 study, with therefore a small number of patients, the study is only statistically powered for the number of patients in each cohort to find differences between occurrence of hypersensitivity or fluid retention. We do describe the occurrence of hyperglycemia, febrile neutropenia and nausea (table 4). Furthermore, in the supplementary material you can find the levels of glucose, insulin and IGF-1 between dose levels and quality of life scores
- Recent clinical trials have demonstrated the benefit of using prednisone or prednisolone, instead of dexamethasone, in conjunction with docetaxel or abiraterone for the treatment of advanced prostate cancer. How would the results presented in this manuscript with a small cohort of patients impact the current treatment of advanced prostate cancer in light of the several recent clinical trials evaluating the benefits of prednisone compared to dexamethasone?
This is an important and relevant finding. Thank you for pointing that out to us. We have included this in the discussion section. Among the other reasons why you may not want to completely omit corticosteroids from docetaxel treatment.
- The recent multi-center studies by Tanaka et al (Int J Urol 2019, Japan J Clin Oncology 2017) in larger patient cohort studied the safety of docetaxel-chemotherapy combined with low-dose dexamethasone (0.5-1mg) and demonstrated excellent safety with a survival benefit for castration-resistant prostate cancer patients. The authors did not compare their results with a reduced dose of dexamethasone (4 mg) to those of Tanaka et al. Further, the studies by Tanaka et al were not cited.
The studies from Tanaka have now been discussed in the discussion section.
- How did the study account for patients using corticosteroid treatment for other conditions while enrolled in this clinical study?
Use of corticosteroid treatment for other conditions was an exclusion criterium for participation in this study, this has been added in the methods section.
Minor points
- Lines 112 – 113 have different font types
Maybe this has been adjusted by the editor, it has the same font types in our uploaded manuscript.
- In table 3, “Number ofevaluablepatients” should indicate "number of patients evaluated"
It has been changed as suggested.
- The link on reference 12 takes to a page that no longer exists
Than you for pointing this out. It has been changed to the correct page.
https://www.ema.europa.eu/en/documents/overview/taxotere-epar-summary-public_en.pdf

Reviewer 3 Report
The article evaluated the safety of reducing the dose of dexamethasone around docetaxel infusion. The results are of great clinical significance. The results show that small doses of dexamethasone can still make docetaxel safe to use. The experimental design is reasonable, the results are reliable, and the quality of the article presentation is also high. It can be accepted.
But there still some small questions.
1. The sample size of this study is very small. Phase I clinical is mainly used to evaluate the safe dose and metabolic effect of the drug. The incidence of allergy to docetaxel is relatively small. The clinical sample size of this study can not yet obtain a relatively positive dose experimental data.
2. Discussion part:
High dose dexamethasone may be associated with severe side effects.
References and specific side effects should be given. The dexamethasone pretreatment of docetaxel is a short-term hormone use, and the real side effects are not unacceptable, compared with the lethal anaphylaxis of docetaxel.
3. In Table 2, the use of punctuation and precision is inconsistent and Line 1 should be replaced by: 69.5 (55-80) 54.0 (34-67)
Author Response
Response to reviewer 3
- The sample size of this study is very small. Phase I clinical is mainly used to evaluate the safe dose and metabolic effect of the drug. The incidence of allergy to docetaxel is relatively small. The clinical sample size of this study can not yet obtain a relatively positive dose experimental data.
You are right that it is a small study. However, the study is powered for the number of patients in each cohort to find differences between occurrence of hypersensitivity or fluid retention. The substantiation for this calculation is in the statistical analysis paragraph. Furthermore, in the discussion we emphasized this small sample size even more to add a more nuance. We already stated in the conclusion that because of the small number of patients, larger studies are required to confirm our results.
- Discussion part:
High dose dexamethasone may be associated with severe side effects.
References and specific side effects should be given. The dexamethasone pretreatment of docetaxel is a short-term hormone use, and the real side effects are not unacceptable, compared with the lethal anaphylaxis of docetaxel.
The potentially side-effects of dexamethasone have been given in the introduction with corresponding references. That was the reason we did not explored this again in the discussion initially. We have now expanded the text in the discussion. The remark on short-term corticosteroid use in respect to risk of side effects has been added in the discussion part
- In Table 2, the use of punctuation and precision is inconsistent and Line 1 should be replaced by: 69.5 (55-80) 54.0 (34-67)
Thank you for pointing out this inconsistence. It has been adapted in the table.

Reviewer 4 Report
Docetaxel is a widely used anti-cancer drug but is associated with high rates of hypersensitivity reactions and fluid retention. To reduce this issue, a high dose of systemic corticosteroids is used in the clinic. However, a high dose of corticosteroids is associated with side effects. The manuscript by Lugtenberg et al. evaluated the safety of reducing the dose of dexamethasone around docetaxel in patients with prostate and breast cancer. The manuscript is well-written and provides useful information for the further evaluation of dexamethasone dosage reduction. I have a few minor comments as described below.
1. Could you please explain why breast cancer and prostate cancer cohorts have different dose level designs (Table 1)? How would the difference affect result interpretations?
2. There is a big variation in the mean cumulative dose of Docetaxel within each cohort (Table 2). Could you please use a table to provide detailed information for each individual?
Author Response
Response to reviewer 4
- Could you please explain why breast cancer and prostate cancer cohorts have different dose level designs (Table 1)? How would the difference affect result interpretations?
There are different dose level designs, because the current prophylactic dosing of dexamethasone also differs for docetaxel treatment in prostate and breast cancer because of different dosages docetaxel and concurrent use of prednisone in prostate cancer. We added this information in the introduction section and in table 1.
- There is a big variation in the mean cumulative dose of Docetaxel within each cohort (Table 2). Could you please use a table to provide detailed information for each individual?
We made a new table with detailed information for each patient that was evaluable for the endpoint. This new table can replace table 3 or can be added in the supplementary materials.
